# Environmentally Friendly, High-Performance Fire Retardant Made from Cellulose and Graphite

**DOI:** 10.3390/polym13152400

**Published:** 2021-07-22

**Authors:** Leandra P. Santos, Douglas S. da Silva, Thais H. Morari, Fernando Galembeck

**Affiliations:** 1Galembetech Consultores e Tecnologia Ltda., Campinas 13080-661, Brazil; leandrapereiradosantos@gmail.com (L.P.S.); thmorari91@hotmail.com (T.H.M.); 2Department of Physical Chemistry, Institute of Chemistry, University of Campinas, Campinas 13083-970, Brazil; dssilva@unicamp.br

**Keywords:** fire retardants, cellulose, exfoliated graphite, thermal conductivity, water-based paint, wood protection, ERG (exfoliated and reassembled graphite)

## Abstract

Many materials and additives perform well as fire retardants and suppressants, but there is an ever-growing list of unfulfilled demands requiring new developments. This work explores the outstanding dispersant and adhesive performances of cellulose to create a new effective fire-retardant: exfoliated and reassembled graphite (ERG). This is a new 2D polyfunctional material formed by drying aqueous dispersions of graphite and cellulose on wood, canvas, and other lignocellulosic materials, thus producing adherent layers that reduce the damage caused by a flame to the substrates. Visual observation, thermal images and surface temperature measurements reveal fast heat transfer away from the flamed spots, suppressing flare formation. Pinewood coated with ERG underwent standard flame resistance tests in an accredited laboratory, reaching the highest possible class for combustible substrates. The fire-retardant performance of ERG derives from its thermal stability in air and from its ability to transfer heat to the environment, by conduction and radiation. This new material may thus lead a new class of flame-retardant coatings based on a hitherto unexplored mechanism for fire retardation and showing several technical advantages: the precursor dispersions are water-based, the raw materials used are commodities, and the production process can be performed on commonly used equipment with minimal waste.

## 1. Introduction

Cellulose molecules are chemically simpler than most polysaccharides and other natural polymers. However, celluloses obtained from different living species display remarkably diverse properties due to their hierarchical self-assembly [1,2]. Moreover, exploiting the polymer’s chemical modification and chirality is increasing the ever-growing list of applications of cellulosic materials rapidly [3,4,5,6,7].

The self-assembly and solubility of cellulose are now better understood, considering its amphiphile character analogous to DNA [8,9] including significant stacking interactions [10]. However, the recognition of cellulose’s amphiphile properties is recent [11].

This group explored cellulose amphiphile properties to create re-pulpable paper adhesives from aqueous cellulose solutions [12]. An attempt to explore stacking interactions between cellulose and planar molecules revealed that cellulose solutions wet graphite powders in alkaline water, dispersing and exfoliating graphite crystals [13], and stabilizing aqueous graphite dispersions. These coat wood, paper, cloth, and non-woven textiles made from natural and synthetic polymers, producing adhesive layers of exfoliated and reassembled graphite (ERG). Transmission electron microscopy (TEM) micrographs of dilute dispersions showed thin graphite particles while scanning probe microscopies and micro-Raman imaging revealed that the dry coatings are formed by extensive layers of superimposed lamellae, joined by small patches of cellulose [13]. The dry coating films are thus wetted by water with 50 degrees contact angles that decrease slowly due to water absorption. Coating films are cohesive and adherent to wood and other materials (class 5 following ASTM D 3359-97 standard). The application of successive dispersion layers on a substrate produces smooth and adherent layers of increasing thickness that resist the adhesion of foreign materials, like adhesive tapes. This is due to graphite exfoliation by peeling that was the method used to make the first isolated graphene sheets [14]. 

These are electric conductors whose surface electrical resistance can be controlled in a very wide range according to the application required, with the current lower limit of 0.5 Ω/square. Higher electrical resistance films (up to 10 kΩ/ square) are obtained by adding low-conductive fillers to the dispersions, as required by the intended application. ERG-coated paper and cloth are thus flexible conductive sheets suitable for making electric circuit components, but this raises safety questions related to their flammability. 

Public concern about fire triggers research activity that has already created effective fire-protective additives for polymers, [15,16,17,18,19,20] like hydrated compounds, for example, hydrated salts [21], oxides (e.g., alumina) and clays that undergo endothermic degradation, carbonates like huntite that decompose forming a CO_2_ gas blanket, halogenated paraffins and polymers that emit free-radical suppressants, and chemicals that intumesce like the expandable ammonium phosphates [22], or still those that form a barrier between air and the substrate, e.g., silica [23,24,25,26] and clays [27,28], or char as, for example, organophosphorus compounds [16,29,30,31], polyols and melamine. Graphite (flakes, powder or expanded) has been used as an additive in intumescing coatings [19,20,32,33,34,35] and in polymer composites [23,24,25,26,27,28,29,30,36,37,38,39,40,41,42,43,44,45]. Thus, existing fire-retardants explore different approaches and mechanisms. 

However, each fire suppressant or retardant presents some limitations due to cost, toxicity, environmental risk, and effectiveness that may prevent its widespread adoption and leaves many old demands often unanswered by current products [46,47,48,49]. Moreover, new demands arise continuously [50,51], spurring a steady research and development activity that is evidenced by hundreds of annual publications and 912 patent applications in the USPTO, since 2001. For this reason, many chemical and materials researchers target new fire retardants and suppressants [52,53].

In this work, we describe the fire-retardant properties of ERG coatings, adding a new property to this polyfunctional material and showing how the dispersant cellulose contributes to build an effective fire barrier. 

## 2. Materials and Methods

### 2.1. Materials

Graphite powder (GrafineR 95100, 96.4%C, 3.5% ash, mesh size ≥325) is a product from Nacional de Grafite (São Paulo, Brazil). Blanver (São Paulo, Brazil) supplied the microcrystalline cellulose (MicrocelR-101, 50-micrometer average particle size, 0.26–0.31 g cm^−3^ density). Synth (Diadema, Brazil) supplied the activated carbon, NaOH solution (50% m/m) and alumina. Other suppliers are: canvas fabric 534 g m^−2^, pine wood planks, 15 mm of thickness and density 0.46 g cm^3^).

### 2.2. Sample Preparation and Characterization

#### 2.2.1. Graphite Dispersion and Rheology Characterization 

Dispersions with different proportions of graphite and cellulose were prepared as described in previous works of the authors [13,14]. Cellulose powder was slowly added to a cold (0–5 °C) NaOH solution in a polyethylene beaker under gentle stirring using a Fisatom 713D stirrer at 210 rpm min^−1^ to avoid cellulose clumping. The resulting dispersion was kept in a refrigerator overnight, and then stirred and examined to ensure that the cellulose was fully dissolved. Cold water was added as required to achieve the desired cellulose concentration. The required amount of graphite was then slowly added to the cellulose solution that was kept under gentle stirring for one hour and stored at a low temperature. This procedure was used for a broad range of concentrations: graphite, from 2% to 20% weight; cellulose, 0.5% to 4% weight; and NaOH, 3% to 7% weight. Depending on the specific application, other additives are added to the graphite dispersion, e.g., activated carbon and alumina.

Considering the large number of possible dispersion compositions obtained by this procedure, the dispersion composition of the samples used to prepare the samples for standard testing was defined using a three-factor two-level fractional factorial design with three central points. Experimental runs are designated as N1-N11 in Table 1. ERG-a to e are different compositions of exfoliated and reassembled graphite. Four responses were evaluated: viscosity, adhesion to wood, flame damaged area, and coating solubility in water.

NaOH concentration was constant, at 7% m/m.

Viscosity was measured using a DV-II + Pro rotational viscometer (Brookfield, Middleboro, MA, USA). A volume of 0.5 ml of each sample was placed between the viscometer plates and shear stress measurements were obtained at different rotation speeds. The rotor used was the CP52 model, which allows viscosity measurements up to 96,000 cP. The samples were kept at 20 °C during the test using a thermostat.

TEM measurements were performed using a Libra 120 instrument(Carl Zeiss NTS GmBH, Oberkochen, Germany) operated at 80 kV accelerating voltage with a tungsten filament. The microscope was equipped with an in-column omega filter. Brightfield images with low chromatic aberration were collected with a 15 eV-width energy slit selected to zero loss. The same configuration was also used to obtain 25 eV energy-loss filtered images (EFTEM). The samples collected in the sedimentation experiments were diluted 1000-fold in water and a droplet of the dilute dispersion was placed over a parlodion-carbon coated TEM microscope grid. All data were registered using an Olympus Cantega G2 CCD camera (2048 × 2048 pixels) controlled by iTEM software version 5.2 (Olympus Soft Imaging Solutions GmbH, Münster, Germany). 

#### 2.2.2. Exfoliated and Reassembled Graphite (ERG)-Coated Samples Preparation and Characterization

Graphite dispersions were applied to pine wood planks and cotton canvas pieces using a synthetic bristle paint brush. Depending on the desired ERG coating weight per substrate unit area successive applications were made prior to the next application. 

Various finishing procedures can be used, depending on the target application, aesthetic and other requirements as well as the limitations posed by the substrate. Calendering produces the best results for canvas, and this was done in a two-roll MH-300C calender mixer (MH equipamentos, Guarulhos, Brazil), while sanding and polishing were suitable for finishing ERG coatings on wood. 

The thicknesses of the solid films formed after drying were measured using a digital micrometer (293-721-30, Mitutoyo, Suzano, Brazil). Electrical sheet resistance was measured using a Fluke 115 multimeter connected to a square contact cell built from an acrylic block holding two parallel copper sheets. The adhesion test was performed according to ASTM D 3359-97 standard. Coating solubility in water was determined by immersing the sample in water for 24 h, filtering and weighing any solids fallen from the sample. The morphology of the solid coatings was evaluated by acquiring surface and cross-sectional micrographs of the samples using a Quanta 650 FEG microscope (FEI, Brno, Czech Republic), using an acceleration voltage of 10 kV. Samples previously fixed in stubs with carbon tapes received a thin layer of carbon deposited by evaporation using a MED-020 Coating system (Bal-Tec AG, Balzers, Liechtenstein).

#### 2.2.3. Flame-Resistance Tests

Preliminary coating performance tests were carried out comparing samples of coated wood with bare wood controls. The experimental set-up used in the laboratory to observe the behavior of coated wood under fire is shown in Figure 1. Similar tests were carried out on cotton canvas samples coated with ERG paint.

The blue flame of a small torch of liquefied petroleum gas (LPG) was applied to a point in the sample, so that the flame cone reached the point chosen on the surface, within one millimeter. The flame was applied for a determined time and then removed, video and thermal images of the samples were obtained, until the time necessary for the flame to extinguish. Thermal images were obtained using a FLIR TG297 high temperature thermographic camera with a spectral range of 7.5 to 14 µm. 

Type K thermocouples were placed on the surface of each sample, a few centimeters away from the point of application of the flame, and electrically connected to a data acquisition system that allows the recording of the temperature at each point, as a function of time.

The preliminary tests were followed by testing in an accredited laboratory, using the standards ASTM E662, NBR 9442 and ISO 1182. Since the results obtained were highly positive (see the Results section and Appendix A), the authors did not perform cone calorimetry and other tests often presented in the literature. 

## 3. Results

### 3.1. Preparation and Properties of Graphite Dispersions

The black-gray graphite dispersions are visually stable, and they do not show the formation of coagula. The particles undergo extensive exfoliation that is evidenced by transmission electron microscopy from particles collected from dilute solutions. 

Representative micrographs are shown in Figure 2, where the two upper images are from the same field but acquired using different imaging modes. Figure 2a is a bright-field picture, where the darker areas correspond to thicker particles. Figure 2b is a low-energy loss image that shows the inner details of the thick particles. This mode is not so frequently seen in the literature, but it has great potential for microanalytical work and for the examination of thick specimens and of very thin layers. 

The two images show overlayed lamellae partly connected by rounded domains and coated by tenuous layers that are assigned to cellulose.

The presence of cellulose and its adhesion to the graphite/graphene lamellae is shown more clearly in Figure 2c,d, where thin cellulose films are bound to graphite lamellae sides and surfaces, forming large holes as well as “necks”. These morphologies are expected for a plasticized non-crystalline polymer drying while adhering to a stiffer material. In the present case, the polymer is cellulose, bound to graphite/graphene. 

The aqueous dispersions are colloidally stable, the particles do not form clumps or aggregates, and neither do they adhere to the walls of polyethylene containers. Slow sediment formation occurs, which is expected due to the high lateral particle dimension in the 1 micrometer range. 

The stability of the dispersions was assessed by measuring their viscosity as a function of time, due to the difficulty posed by using optical methods in very dark dispersions. Results of viscosity measurements of various dispersions with different concentrations, and within a broad shear rate range, are presented in Figure 3. Dispersions containing 2% (m/m) of cellulose, 10% or 15 % (m/m) of graphite and 2% or 5% (m/m) of activated carbon show Newtonian behavior, but the more concentrated ones are non-Newtonian, and they are thinned under higher shear rates. The overall viscosity range covers four orders of magnitude, which shows the possibility of applying dispersions by using various coating techniques, at different speeds.

The pronounced dispersion thinning is easily understood considering the graphite particle anisometry, comparable to that shown by montmorillonite clay and the lamellae formed in concentrated surfactant solutions. Graphite anisometry also produces an interesting rheo-optical effect: the resting dispersions are very dark, but their surfaces shine under stirring.

At still higher concentrations than those used in Figure 3, the rheological behavior becomes much more complex, including the formation of gel at dilatant behavior. 

The viscosity of the samples presents small variations with the storage time, which confirms their colloidal stability. Viscosity measurements are thus a convenient way to monitor dispersion stability, in these dark dispersions that are not amenable to study by optical methods.

### 3.2. Coating Characteristics

Microscopic examination (Figure 4) of films adherent to wood shows (i) the coexistence of graphite and cellulose, (ii) the accumulation of cellulose at the interface between graphite and substrate and (iii) the horizontal alignment of the reorganized graphite lamellae, which is responsible for the high electrical and thermal conductivities of the coatings.

The results of the factorial experiment defined sample ERG-a as the best candidate for further experiments on flame resistance. ERG-a dispersion produces a coating strongly adherent to wood, insoluble in water (solid particles leached in water < 0.1%) and showing the highest resistance to flame. The sheet resistance of the ERG-a coating applied to wood varies from 265 to 90 Ω/square when the thickness varies from 100 to 250 μm. 

Figure 5 shows the yield (coated area per dispersion unit weight) of the ERG-a dispersion as a function of the thickness of the dry coating. Different thicknesses are easily obtained by diluting the ERG-a dispersion (the first three points (blue or red) from left to right of Figure 4 were obtained diluting the ERG-a dispersion 2.5-fold, 2-fold and 1.5-fold) or by applying successive coats (1 to 6 layers, from the fourth point (blue or red) from the left to the rightmost point). Preliminary tests in the laboratory indicated that flame propagation protection is obtained when the coating thickness exceeds 250 μm.

The best results on canvas were obtained using an ERG dispersion whose weight composition was 4% cellulose, 7% NaOH and 20% graphite. The as-prepared coating is 70 μm thickness, lowering to 50 μm after calendering. The finishing procedure has a positive impact on sheet resistance which reduces from 84–150 Ω/square to 3–6 Ω/square.

### 3.3. Exposure to Flame: Laboratory Tests

The protocol for the wood assay used in these experiments is presented in Figure 6. Temperature measurements at a point adjacent to the flame application were made with the optical thermometer of the thermal chamber.

Figure 7 shows pictures of samples assembled as in Figure 1, following the protocol described in Figure 6.

Flame applied to uncoated wood produced a flare and drilled a hole through wood. The coated samples did not flare, the plates were not pierced, and the back side was unchanged. The readings of the thermocouples and the optical thermometer are presented in Figure 8. Photos obtained with the thermal camera are shown in Figure 9.

The figures show that the temperature rises very rapidly in unprotected wood and in wood with ERG, but more slowly in wood coated with ERG + alumina, which is explained by the endothermy of the alumina dehydration reaction. However, after two minutes’ exposure, the additional protective effect of alumina is no longer perceived. In coated wood, the temperatures measured outside the point of flame application are much higher than the ambient temperature, while in uncoated wood it is always just above the environment. This demonstrates the effect of the high thermal conductivity of graphite, which dissipates the thermal energy received from the wood by the flame, preventing the gasification of the wood and the inflammation of the gases produced. The high thermal conductivity of graphite also explains the faster decrease of the temperature, after the flame’s contact with the coated wood samples is interrupted. 

Canvas made from raw cotton was also tested, using the same experimental arrangement and protocol used in the wood test but the samples were exposed to flame for 1 min, only. Images recorded from the tested canvases are in Figure 10. Both samples burn forming a hole, but this effect is remarkably delayed in the coated tissue. The fire ends in the coated sample, while an ember persists on raw canvas, increasing the hole in the fabric for at least 14 min after the torch flame was withdrawn. The ember was extinguished only by cooling with water.

Thermocouple readings and thermal images obtained during this test are in Figure 11 and Figure 12. 

The steep temperature changes in the untreated cloth are expected since the fast burning in the flamed spot produces a hole lined with glowing coal that propagates radially, enlarging the hole. On the other hand, the flamed spot in the ERG-treated cloth never reaches the same high temperatures, thanks to the fast heat transfer along the protective coating. Heat conduction also explains that the damage stops soon after the flame is withdrawn due to fast heat transfer from the burnt area.

### 3.4. Exposure to Flame: Standard Tests 

The results of standard testing undertaken at IPT (São Paulo, Brazil) are reported in full in the Appendix A, because IPT does not authorize the partial reproduction of their reports. The conclusion is based on Table 1 of section “3—RESULTADOS DE ENSAIO” (Test Results) of Appendix A and the classification of the material is based on section “4—LIMITES ESPECIFICADOS EM NORMA” of the same annex. In short, it places the wood coated with ERG-based fire-retardant in Class IIA, the highest possible for a combustible material. Moreover, the values obtained for the Surface Flame Propagation Index (I_p_) and for the Specific Optical Smoke Density (D_m_) are well below the upper limits of the parameters that define the material class.

## 4. Discussion

Many factors contribute to the performance of ERG coatings as barriers to fire propagation beyond those shown by the currently used graphite fire retardants. ERG is a macroscopic 2-D, high aspect ratio cohesive film with a <20% m/m cellulose content that is its only combustible component.

The experimental results in this paper, e.g., Figure 7, demonstrate the fast heat transfer away from the flamed surface when this is coated with ERG. This is also seen in Figure 11, where a spot adjacent to the flamed area does not reach the same high temperatures as the uncoated surface. Moreover, the coated surface cools down much faster than the uncoated surface, after the flame is withdrawn. Indeed, the thermal images are intrinsic demonstrations of heat radiation. Preventing high temperatures in the substrate is essential to slow its pyrolysis that emits flammable gases, which feed the flame. Thermal conductivity data for ERG have not yet been determined and this is experimentally more difficult than measuring electrical conductivity, largely due to the multiplicity of paths for heat transfer through the substrate and the surrounding atmosphere. Also, graphite is an effective IR radiation emitter. Acquiring heat transfer coefficients for ERG is important but it goes well beyond the scope of this paper. Nevertheless, recalling the basic ideas on electric and thermal conductivity allows us to expect that thermal conduction in the ERG film depends on structural features that also determine its excellent electrical conduction.

Another important contribution to the protective role of ERG coatings is the film morphology shown in the SEM micrographs. The flat reassembled graphite particles are effective barriers for oxygen access to the film that can only take place through the interstices filled with cellulose. Moreover, graphite is anisotropic and its thermal conductivity along the molecular plane is nearly 400 times higher than normal to it, at 300 K and at 900 K [54].

The ability to change the electrical and thermal properties of a material by controlling its morphology opens interesting possibilities for the development of new smart materials and devices, as in the creation of shape-memory nanocomposites [55].

In this work, cellulose accumulates the roles of wetting agent, graphite dispersant, and exfoliant in the aqueous dispersions, also performing as an adhesive and cohesive agent in the dry coating. This effectiveness derives from the unique structural features of cellulose amphiphile macromolecules that confer their ability to bind to graphite and graphene surfaces. This has been recently further verified by making concentrated graphene dispersions [56].

The adhesive property of cellulose imparts mechanical advantage to the coatings. Successive layers of dispersion applied to the substrate increase the thickness of the coating without harming its cohesion. This characteristic also allows minor fire damage to the coating to be repaired easily. Furthermore, the protective ERG coating thicknesses used in this work are lower than those observed by the authors in commercial product labels.

Thickness also affects other functional properties of ERG films; the most remarkable case is its impact on the surface electric resistance with a surprising consequence: ERG coatings may show a surface resistance lower than graphene monolayers the resistivity of which is only 10^−8^ Ω·m^−1^, but the <1 nm thickness leads to a >30 Ω/square [14].

Applying ERG to practical applications does not reveal any significant problems: the precursor aqueous dispersions are conveniently used as paint, alone or compounded with other water-compatible or soluble components. The raw materials are abundant commodities: cellulose is a natural polymer, and graphite is a widespread mineral that can also originate from renewable carbon sources, including garbage [57]. Making and applying the dispersions is undertaken using standard equipment in the paint industry, producing minimal waste under mild conditions. The authors are now working to scale up ERG production and application with prospective users.

Finally, exploring the versatility of soluble cellulose demonstrated in this and previous work will probably create other multifunctional soft materials.

## 5. Conclusions

New flame-retardant coatings made from exfoliated and reassembled graphite (ERG) show excellent performance in protecting wood and other polymer materials from fire. The graphite exfoliant is soluble cellulose that also performs as a cohesive and adhesive agent, thanks to the adhesive properties of cellulose, thus conferring excellent substrate protection against fire. The combination of cellulose adhesion and dispersing properties is another demonstration of the versatility of this polymer due to its recently acknowledged amphiphile character.

Fire protection by 2D ERG coatings is explained by their ability to transfer heat away from the areas exposed to a flame. Other essential contributions comes from the intrinsic resistance of graphite to oxidation under air, up to 600 °C, and from the oxygen barrier properties of the flat lamellae horizontally aligned with the surface. 

ERG is made from commodities using standard processes in water-based polymer dispersion production and applications, e.g., in the paint industry. It is a zero-VOC product whose fabrication leaves minimal amounts of residues that are easily composted or recycled. These advantages should contribute to its widespread application.

## 6. Patents

A patent application was filed in the Brazilian patent office, INPI.

## Figures and Tables

**Figure 1 polymers-13-02400-f001:**
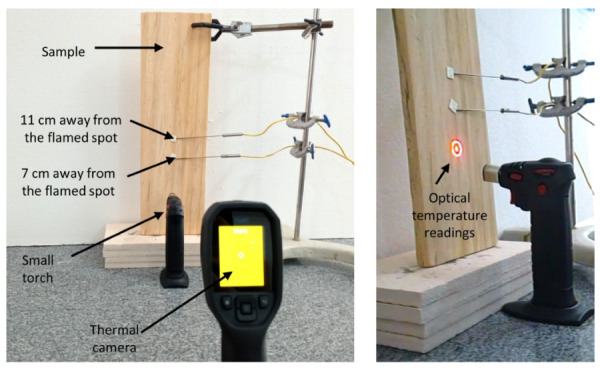
Experimental set-up used to acquire thermal images of wood following its exposure to fire, while measuring the surface temperature at two positions away from the flamed spot.

**Figure 2 polymers-13-02400-f002:**
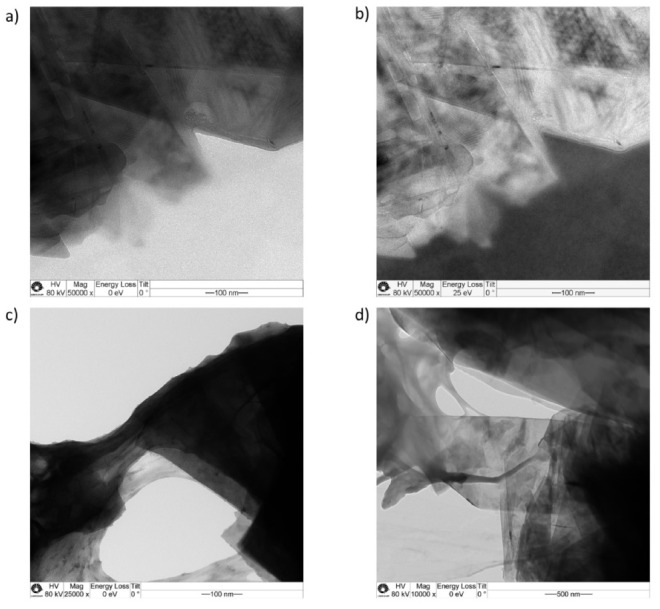
Transmission electron micrographs of dried dispersions of cellulose and graphite. Images (**a**,**c**,**d**) are bright-field micrographs of different areas while (**b**) is a low energy loss image of the same area as (**a**). The scale bars and microscope operation parameters are in the bottom line of each micrograph.

**Figure 3 polymers-13-02400-f003:**
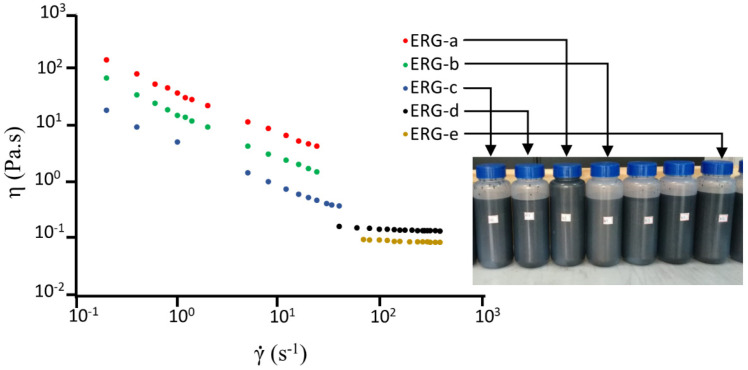
Viscosity (η) versus shear rate ($\dot{\gamma }$) plots of five different dispersions of graphite exfoliated within aqueous cellulose solutions. The photo shows dispersion containers that were kept undisturbed for 16 h, showing the absence of clear supernatant.

**Figure 4 polymers-13-02400-f004:**
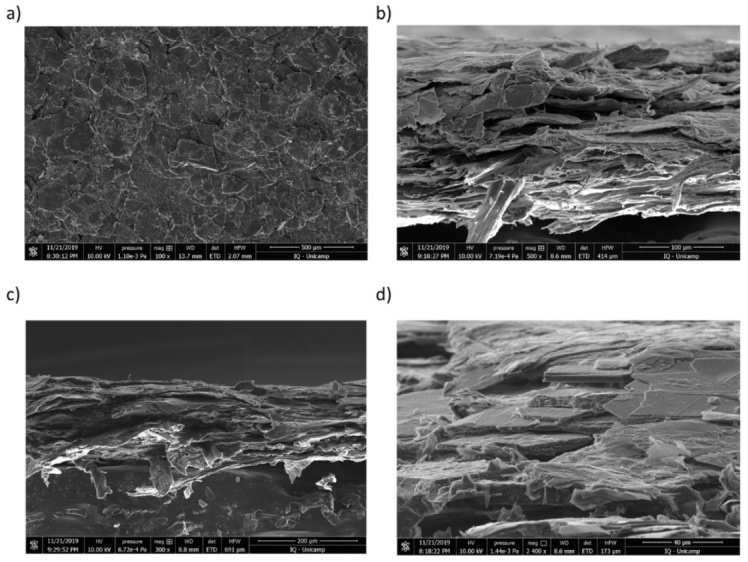
Electron scanning micrographs of ERG coatings applied on the substrate: (**a**) surface, (**b**,**c**) cross-sectional and (**d**) tilted views of the specimen. The scale bars and microscope operation parameters are in the bottom line of each micrograph.

**Figure 5 polymers-13-02400-f005:**
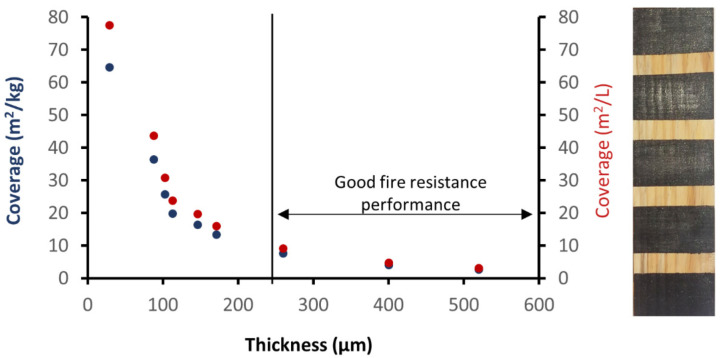
Thickness of coating layers vs. the coated area/paint weight or volume used. The photo on the right shows the result of ink application tests, using different dilution factors or number of paint applications. Note the good coverage and the absence of runoff or brush marks.

**Figure 6 polymers-13-02400-f006:**
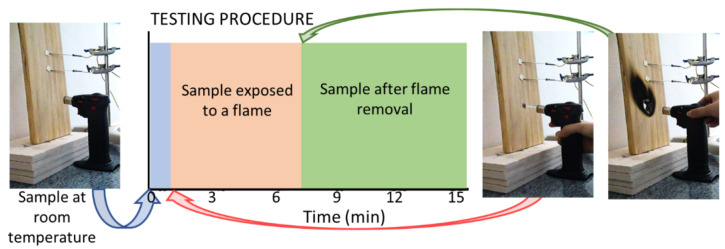
Test protocol to assess coating resistance to flame: the samples were observed for 1 min at room temperature, the flame was applied for 6 min and then withdrawn. Sample observation continued for another 8 min.

**Figure 7 polymers-13-02400-f007:**
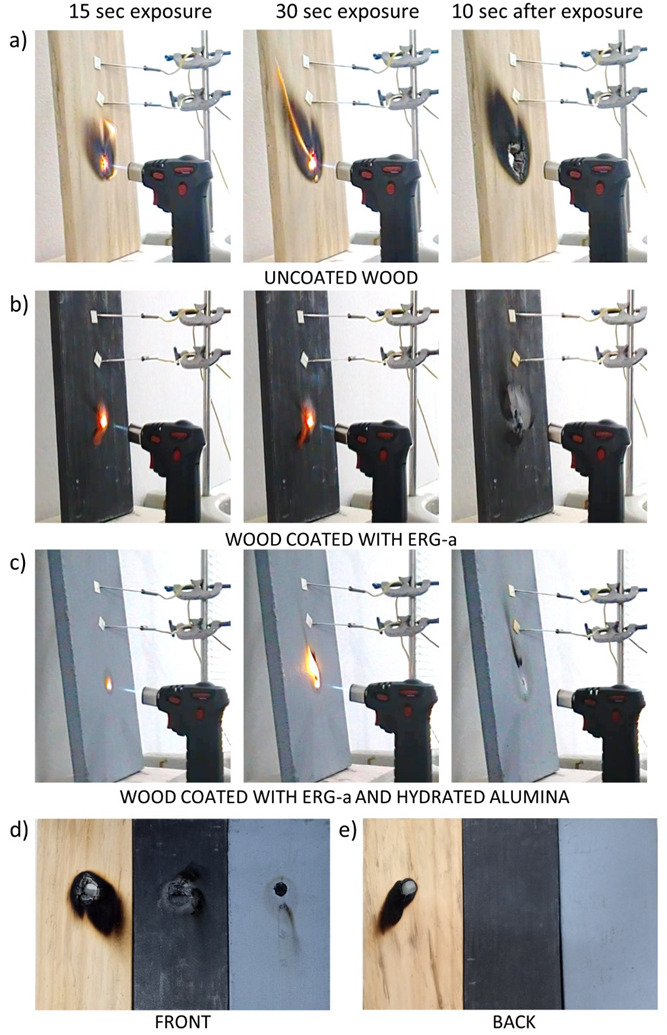
Photos of uncoated wood (**a**) and wood coated with ERG-a (**b**) and ERG + hydrated alumina (**c**) at the times indicated during their exposure to the flame and after the flame removal. The two bottom pictures show the three samples after the test (**d**,**e**), allowing the visual assessment of the damage suffered.

**Figure 8 polymers-13-02400-f008:**
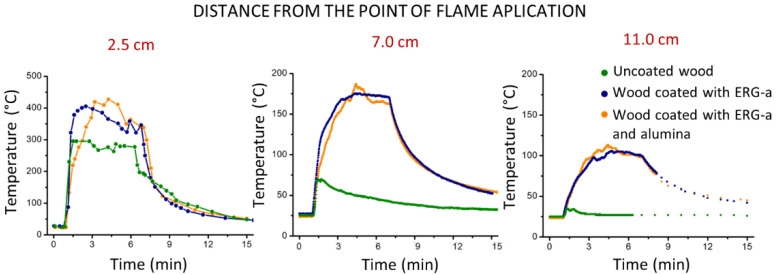
Temperature vs. time change in three positions on the wood sample. The readings at 2.5 cm were obtained using the optical thermometer, while the others were obtained using the thermocouples. The readings with the optical thermometer were not corrected for the materials emissivity.

**Figure 9 polymers-13-02400-f009:**
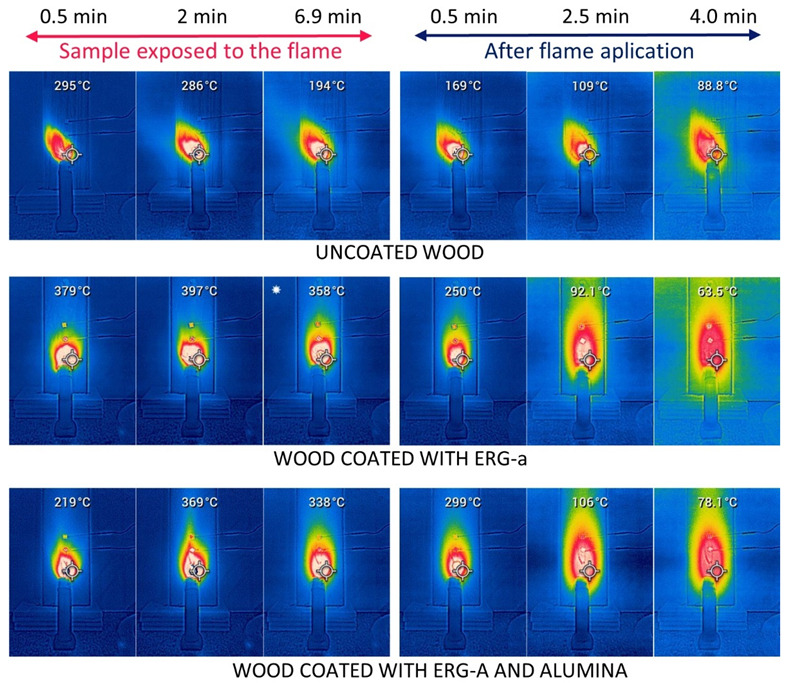
Thermal photos of the surface of untreated wood and wood coated with ERG-a or ERG + hydrated alumina. Temperatures are not corrected for emissivity.

**Figure 10 polymers-13-02400-f010:**
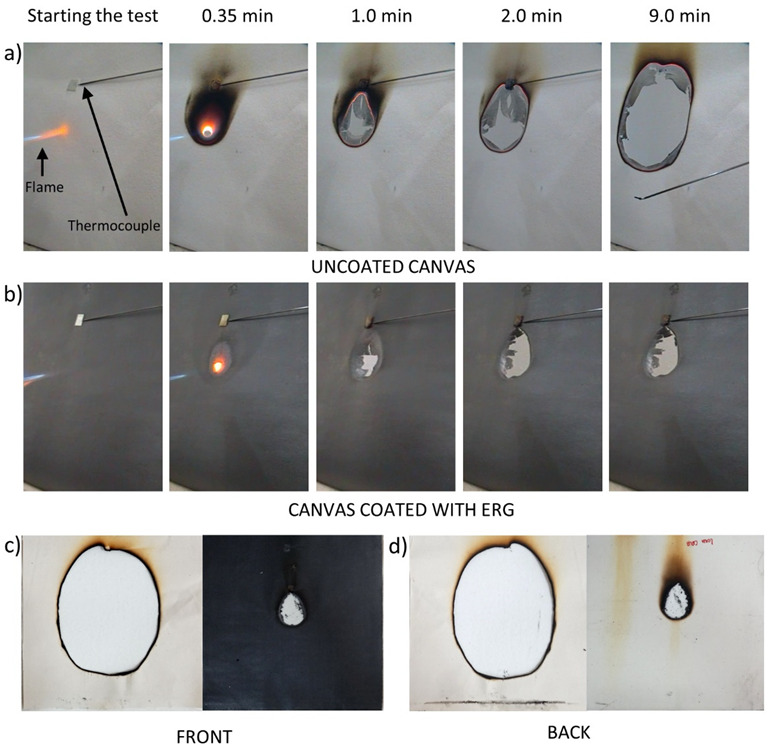
Photos of uncoated raw cotton canvas (**a**) and canvas coated with ERG (**b**) exposed to flame. Note that the hole growth in the coated canvas stops 2 min after removing the flame, but it continues indefinitely in the uncoated canvas. The last photo shows the two samples after the test (**c**,**d**).

**Figure 11 polymers-13-02400-f011:**
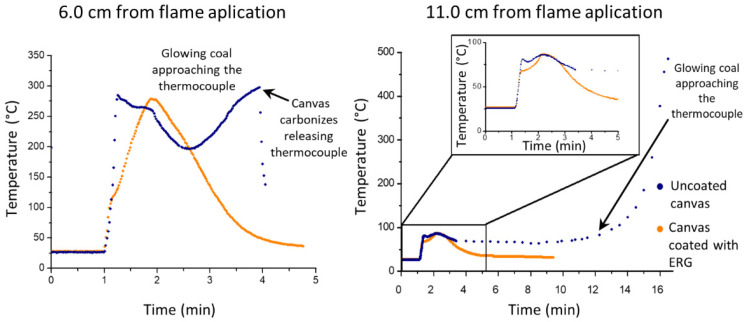
Temperature vs. time variation in two positions distant from the flamed spot.

**Figure 12 polymers-13-02400-f012:**
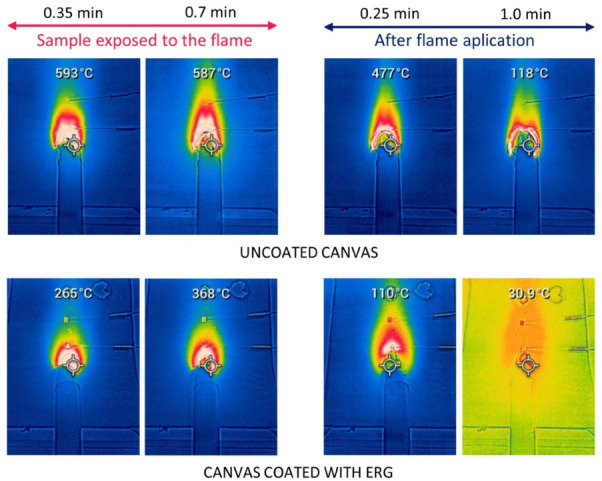
Thermal photos of the surface of untreated raw cotton canvas and ERG-coated canvas. The values shown at the top of the images correspond to the optical readings at 2 cm distance from the flame’s point of incidence.

**Table 1 polymers-13-02400-t001:** Sample composition in the factorial design runs.

Run/Sample Designation	Weight Percentage (% m/m)		
	Cellulose	Graphite	Activated Carbon
N11	3	12.5	3.5
N1	2	10	2
N5	2	10	5
N3/ERG-e	2	15	2
N4/ERG-b	4	15	2
N8/ERG-a	4	15	5
N7/ERG-d	2	15	5
N6/ERG-c	4	10	5
N10	3	12.5	3.5
N2	4	10	2
N9	3	12.5	3.5

## Data Availability

The data presented in this study are available on request from the corresponding author.

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
