# Peer review of "Environmentally Friendly, High-Performance Fire Retardant Made from Cellulose and Graphite"

_polymers, 2021, doi:10.3390/polym13152400_

Round 1

Reviewer 1 Report

In this work, the authors described the fire-retardant properties of coatings based on exfoliated  and reassembled graphite, and was introduced as an electrically conductive coating for wood and paper. This manuscript was well written. Therefore, I suggest accepting this paper after a major revision. Before resubmitting, the authors should solve the following problems:

  1. The title of 1.1 Fire and safety should be deleted.
  2. The introduction of background is not enough, in addition, the authors should emphasize the novelty of this work.
  3. The part content of Discussion should be transferred in introduction part,
  4. In most cases, the authors just described the results of experiment, the retardant mechanism should be discussed in more detail.
  5. Part 3.1 “The particles undergo extensive exfoliation that is evidenced by transmission electron microscopy from particles collected from dilute solutions.”, the authors should added these TEM images in this part.
  6. Part 3.2 “Electrical resistance of the coating film that can be controlled in a very wide range, with the current lower limit of 0.5 Ω/square.”. the authors should describe the conductivity of the coating film more accurately, this range is too large.
  7. Also, the authors should describe the contact angles of the coating film more accurately.
  8. The authors should give the thermal conductivity of the coating film.
  9. More instruments such as XRD should be used to characterize the fire retardant.

Reviewer 2 Report

This study reports on environmentally friendly, high performance fire retardant made from cellulose and graphite, and a series experimental results and theoretical analysis have been presented to support it. After carefully reading it, I think I would like to recommend it after the following revisions are considered.

1. “and related materials” in ABSTRACT, please clearly claim what they are.

2. “1.1. Fire and Safety” is suggested to remove, and furthermore, the research on cellulose and graphite should be presented in INTRODUCTION, and help the readers to catch up with the research history on them.

3. Curve is suggested to use for the data in Figure 2.

4. Discussion on the dispersion of graphite on wood is not enough in section of 3.2, such as alignment of the filler, the thermal conductivity should be significantly influenced, and the working mechanism behind it is not clearly presented, the following references may be useful for it, Haibao Lu et.al., Nanopaper Enabled Shape-Memory Nanocomposite with Vertically Aligned Nickel Nanostrand: Controlled Synthesis and Electrical Actuation. Soft Matter. 2011, 7(16): 7416-7423.

5. “Graphite is anisotropic and its thermal conductivity along the molecular plane is 47 times higher than normal to it, at 300 K [39].” In section of 4. Discussion, how about the conductivity at 600oC? This sentence means the thermal conductivity along surface is higher than that is perpendicular to the surface?

In all, an interesting and useful study.

Reviewer 3 Report

Indeed, an interesting work, the authors have carried all the necessary characterizations. My comments are below;

  1. The introduction part has skipped latest works on same topic, which seems extensively carried out. Please add literature review on earlier similar work.
  2. The methodology part is confusing, I presume due to the patent premises, however more details would be essential.
  3. The thickness of coating should be discussed in detail.
  4. What could be the role of cellulose along with the graphite and adhesives? Cellulosic nature, hydrophilic itself is difficult to disperse, add comments in that. The graphite may assist in dispersion ?
  5. How the authors deduce, its VOC free flame retardant ?

Round 2

Reviewer 1 Report

The author solved all the problems.

Reviewer 3 Report

I have no more comments to add.